# Piperacillin–Tazobactam as an Adjuvant in the Mechanical Treatment of Patients with Periodontitis: A Randomized Clinical Study

**DOI:** 10.3390/antibiotics11121689

**Published:** 2022-11-23

**Authors:** Dolores Hurtado-Celotti, Natalia Martínez-Rodríguez, Pedro Luis Ruiz-Sáenz, Cristina Barona-Dorado, Juan Santos-Marino, José María Martínez-González

**Affiliations:** 1Department of Dental Clinical Specialties, Faculty of Dentistry, Complutense University of Madrid, 28040 Madrid, Spain; 2Surgical and Implant Therapies in the Oral Cavity Research Group, Complutense University of Madrid, 28040 Madrid, Spain; 3Department of Dentistry, Central Hospital of the Red Cross of Madrid, 28003 Madrid, Spain; 4Department of Surgery, Faculty of Medicine, University of Salamanca, 37007 Salamanca, Spain

**Keywords:** dentistry, local antibiotic, piperacillin–tazobactam, periodontitis, mechanical treatment, scaling and root planing, periodontology, single-blind split-mouth randomized study

## Abstract

In this study, the aim was to evaluate the effects of the adjuvant piperacillin–tazobactam solution in the mechanical treatment of periodontitis. A single-blind split-mouth randomized study, it included 24 participants. All of them presented periodontitis stage III according to the 2018 World Workshop classification and the presence of at least one of the following periodontal pathogens: *Aggregatibacter actinomycetemcomitans*; *Porphyromona gingivalis*; *Treponema denticola*; *Tannerella forsythia*; *Prevotella intermedia*. The study established two groups: a control group (SRP: scaling and root planing) and a test group (SRP plus local piperacillin–tazobactam). The final recruitment included 11 women (45.8%) and 13 men (54.2%). The age range was between 25 and 72 years, and the mean age was 57 ± 10.20 years. Clinical controls were performed at 2 weeks, 3 months, and 6 months, repeating the SRP and applying the piperacillin–tazobactam solution again at the 3-month appointment. The clinical attachment level decreased by a mean of 2.13 ± 0.17 mm from the baseline to 6 months in the test group versus 1.63 ± 0.18 mm in the control group. The mean probing pocket depth decreased from 1.32 ± 0.09 mm in the test group, versus from 0.96 ± 0.14 mm on the control side. The plaque index in the test group decreased by 0.46 ± 0.04, while it decreased by an average of 0.31 ± 0.04 in the control group. In conclusion, the local use of piperacillin–tazobactam as complementary therapy produces better clinical results in patients with periodontitis. However, these results are not maintained over time, and so a more persistent local application is necessary.

## 1. Introduction

Periodontitis is a multifactorial chronic inflammatory disease that produces progressive destruction of the support of the teeth. The new classification proposed in 2018 at the World Workshop on the Classification of Periodontal and Peri-Implant Diseases and Conditions means that this disease can be considered under different therapeutic approaches [1].

To remove calculus and biofilm from the tooth surface, nonsurgical mechanical treatment (by professional tooth cleaning and scaling and root planing) is the indicated treatment in patients with periodontitis.

However, the use of this treatment as a single therapy does not always achieve control of the disease, which is why different complementary therapies are used, such as the Nd:YAG laser and the application of different agents (such as sodium hypochlorite, povidone–iodine, or chlorhexidine), which have different results in terms of the clinical attachment level and bacterial load [2].

On the one hand, systemic antibiotic therapy in combination with SRP (scaling and root planing) has shown greater efficacy against the presence of periodontal pathogenic bacteria compared with isolated mechanical therapy on aggressive or recurrent periodontitis, as shown by the meta-analysis carried out by Keestra et al. [3].

However, the problem of adverse side effects and especially a seemingly ever-increasing risk of bacterial resistance urge clinicians to balance the risks and benefits well with each individual patient

Given this improvement, and due to the global trend of reductions in systemic oral antibiotic therapy, researchers are investigating the use of local antibiotics with the aim of reducing the probing pocket depth and increasing the clinical attachment level compared with individual mechanical treatment [4,5,6].

Researchers have used different administrations of different antibiotics such as metronidazole gel, tetracycline fibers, doxycycline gel, minocycline gel or microspheres, azithromycin gel, and clarithromycin gel in different fields of dentistry [7,8,9,10,11,12,13,14,15]. In the published studies, the researchers observed substantial heterogeneity in terms of the administration protocol and the results obtained.

The combination of piperacillin–tazobactam is frequently used in the medical field; however, there are few studies in the dental field. One of them, conducted by Zirk et al. [16] on odontogenic sinusitis, highlights its much higher effectiveness (93.3%) compared with other antibiotics, such as ampicillin–sulbactam (80%), moxifloxacin (86.3%), and clindamycin (50%).

Researchers have also proven the effectiveness of the use of piperacillin–tazobactam in periodontitis. In a single study published in 2013, Lauenstein et al. [17] demonstrated substantial reductions in *Treponema denticola*, *Fusobacterium nucleatum* spp. *Polymorphum*, *Parvimona micra*, and *Fusobacterium periodonticum* compared with the conventional SRP treatment.

Callow and Martínez-González [18] conducted a study on the inhibitory capacity of different antibiotics against periodontopathogenic germs, demonstrating the effectiveness of piperacillin–tazobactam in vitro. According to the results, piperacillin–tazobactam had a better response compared with that of amoxicillin/clavulanic acid and minocycline.

The lack of studies on this antibiotic association does not allow for a demonstration as to whether its application can be of benefit in this highly prevalent disease. In this context, the aim of the present study was to evaluate the effects of the piperacillin–tazobactam solution as an adjuvant in the mechanical treatment of periodontitis.

## 2. Materials and Methods

### 2.1. Patient Selection

This single-blind randomized study followed the model proposed in the Consort Declaration for clinical trials. The sample size included patients who were likely to be participants in the study and signed the informed consent. The study was approved by the Ethics Committee of the Hospital Clínico Universitario San Carlos de Madrid (CI 19/165-E).

We performed the nonprobabilistic sampling of consecutive cases of patients who attended the Department of Dental Clinical Specialties of the Faculty of Dentistry of the Complutense University of Madrid.

The inclusion criteria were as follows:Patients older than 18 years;ASA I or II patients;Patients diagnosed with periodontitis stage III. According to the 2018 World Workshop classification [1], (Table 1).Patients who had a bacterial load of at least one of the periodontal pathogens at the time of diagnosis.

Patients were excluded from the study if they met one or more of the following criteria:Patients younger than 18 years;ASA III and IV patients;Patients who had received periodontal treatment in the last 6 months;Smoking patients;Pregnant or lactating women;Immunocompromised patients or patients in treatment with bisphosphonates.

### 2.2. Timeline

On the first visit, a complete periodontal examination and completed periodontal chart were performed. Subgingival samples were taken to confirm the presence of periodontopathogens. Once the inclusion criteria were met, and after obtaining informed consent, the SRP was performed on both groups. In the test group, the piperacillin–tazobactam gel solution was applied.

Clinical controls were carried out after two weeks, and then the treatment was repeated three and six months later.

### 2.3. Evaluation of Microbiological Data

Using sterile paper points, a sample of the subgingival fluid from each patient was taken for one minute from one tooth on each side of the mouth, selecting locations with periodontal pockets > 3 mm (Figure 1). The periodontopathogen determination test (PerioPOC^®^, Genspeed Biotech, Rainbach im Mühlkreis, Austria) was evaluated for the presence of five microorganisms that are frequently related to the appearance and progression of periodontitis and peri-implantitis: *Aggregatibacter actinomycetemcomitans* (Aa); *Porphyromona gingivalis* (Pg); *Treponema denticola* (Td); *Tannerella forsythia* (Tf); *Prevotella intermedia* (Pi). The presence or absence of the germs were assessed by taking samples of subgingival plaque with a paper strut from the kit itself in 20 min [19] (Figure 2).

### 2.4. Intervention and Randomization

Eligible participants were randomly assigned to one of two groups (test or control) by an independent researcher (J.M.M.-G.) using opaque envelopes.

In order to determine the sample size, a pilot study was performed, based on the parameter of probing depth. This study was carried enrolling 5 patients with a split-mouth design, obtaining a mean reduction in probing depth of 1.06 ± 0.78 mm and 0.85 ± 0.66 mm, in the test and control sides, respectively.

Using G Power 3.1 (Dusseldorf, Germany), considering an alpha-type error of 5% and a beta-type error of 5%, the estimation resulted in 20 patients per group.

All treatments were carried out under local anesthesia using infiltrative techniques with articaine 4% 1:200,000 (Ultracain^®^, Normon Laboratories, Madrid, Spain).

The mechanical treatment was performed by the same operator without time restrictions and consisted of professional tooth cleaning with ultrasound (Cavitron^®^, Dentsply Professional, York, PA, USA) and scaling and root planing using curettes ((Hu-Friedy^®^, Chicago, IL, USA) on both sides of the mouth with the purpose of removing supra- and subgingival plaque and calculus.

The piperacillin–tazobactam 100/12.5 mg solution (Gelcide^®^, MedTechDental, Allschwill, Switzerland) was used only in the test group (Figure 3).

Gelcide^®^ contains a powder (500 mg of piperacillin and tazobactam) and a liquid (5 mL of hydroalcoholic solution). Both components were mixed just before application and were administered at 2–3 drops per tooth with a different disposable blunt-tip syringe for each patient. Oral hygiene instructions were provided to the patients during the study. No systemic antibiotics or other local adjuvant agents were administered.

### 2.5. Evaluation of Clinical Data

Evaluation of clinical parameters:-Clinical attachment level (CAL): The CAL is the distance from the cementoenamel junction to the bottom of the subgingival sulcus. The measurements were obtained at 6 points on each tooth (mesial, middle, and distal on the buccal; mesial, middle, and distal on the palatal/lingual) in millimeters using a CP12 periodontal probe;-Probing pocket depth (PPD): The PPD is the distance from the gingival margin to the bottom of the subgingival sulcus. The measures were obtained at 6 points on each tooth (mesial, middle, and distal on the buccal; mesial, middle, and distal on the palatal/lingual) in millimeters using a CP12 periodontal probe;-Löe-Silness plaque index (IPL): All teeth were assessed in 4 gingival units (buccal, palatal/lingual, mesial, and distal), assigning a code to each of them. The index value was calculated by adding the numerical value of each gingival unit and dividing it by the number of units studied:0 = No plaque in the gingival area;1 = A thin film of plaque adhering to the free gingival margin and to the adjacent area of the tooth, which could only be recognized by passing a probe through the tooth surface or revealing it;2 = Moderate accumulation of soft deposits within the gingival pocket, on the gingival margin, and/or adjacent to the tooth surface, which was recognizable at a glance;3 = Abundance of soft 1–2 mm-thick material from the gingival pocket and/or on the gingival margin and adjacent tooth surface;-Löe -Silness gingival index (GI): Each tooth was divided into 4 gingival units (buccal, palatal/lingual, distal, and mesial). Each gingival unit was scored from 0 to 3. We assessed the average of all the values obtained:0 = Normal gingiva, no swelling, no discoloration, no bleeding;1 = Mild swelling, slight color change, slight edema, no bleeding on probing;2 = Moderate swelling, redness, edema, bleeding on probing and pressure;3 = Marked inflammation, marked redness, edema, ulceration, spontaneous bleeding, eventual ulceration.

### 2.6. Statistical Analysis

Statistical analysis was performed with the statistical program IBM SPSS Statistics for Windows (Version 27.0, IBM Corp., Armonk, NY, USA) to record detailed descriptions of the data with frequencies and percentages.

The Shapiro–Wilk test was applied to verify the homogeneous variances and normal distribution of the sample. The analysis of variance was used for the repeated measures, Bonferroni adjustment for the intragroup analysis of the clinical variables, and Student’s t-test for the paired samples to compare the intergroup differences in the clinical attachment level, probing depth, plaque index, and gingival index. The McNemar test was applied to compare the presence of the bacteria studied between and within the groups at different times of the study. The study sample presented a confidence level of 95%. A *p*-value ≤ 0.05 was considered statically significant.

## 3. Results

### 3.1. Study Sample

Patients were recruited from the Department of Dental Clinical Specialties of the Faculty of Dentistry of the Complutense University of Madrid. Of the 43 evaluated patients, 6 of them fit neither of the inclusion criteria and 11 refused to participate in the study. Therefore, 26 participants were included in the study, who underwent the treatment using the approach described in the methodology. Throughout the follow-up, 2 patients did not complete the study due to their lack of attendance at the control reviews and were excluded (Figure 4). Finally, 24 participants were included in the analysis; 11 of them were female (45.8%), and 13 were male (54.2%). The age range was between 25 and 72 years, and the mean age was 57 ± 10.20 years.

### 3.2. Clinical Results

The means and standard deviations of the clinical characteristics were analyzed at the baseline, 2 weeks, 3 months, and 6 months in Table 2.

The clinical attachment level decreased by a mean of 2.13 ± 0.17 mm from baseline to 6 months in the test group, versus 1.63 ± 0.18 mm in the control group. In both groups, a significant reduction (*p* < 0.05) was reported between the baseline and the other three study times. We observed significant differences with more favorable behavior for the test group in the clinical attachment level at 3 months (*p* = 0.001) and 6 months (*p* = 0.002).

The mean probing pocket depth decreased from 1.32 ± 0.09 mm in the test group versus 0.96 ± 0.14 mm in the control group. Both groups showed a significant reduction (*p* < 0.05) between the baseline and the other three study times. We observed significant differences with more favorable behavior for the test group in the probing depth at 3 months (*p* = 0.024) and 6 months (*p* = 0.004).

The plaque index in the test group decreased by 0.46 ± 0.04, while in the control group, it decreased by a mean of 0.31 ± 0.04. Both groups presented a significant reduction (*p* < 0.05) between the baseline and the other three study times. We observed significant differences with more favorable behavior for the test group in the plaque index at 2 weeks (*p* = 0.000), 3 months (*p* = 0.002), and 6 months (*p* = 0.000)

Finally, the gingival index was reduced by a mean of 0.40 ± 0.04 in the test group, compared with a reduction of 0.27 ± 0.03 in the control group. Both groups showed a significant reduction (*p* < 0.05) between the baseline and the other follow-ups. We observed significant differences with more favorable behavior for the study group in the gingival index at 6 months (*p* = 0.034).

### 3.3. Microbiological Results

Regarding the presence of the five periodontopathogenic microorganisms detected, the presences of Pg, Pi, and Td were higher, with percentages between 58.3 and 62.5% in both groups; while the detections of Tf and Aa were 37.5% and 20.8%, respectively (Table 3).

In the control group, a significant reduction in the microorganisms at the 2-week visit was observed, which produced new colonization at three months. After performing a second treatment at 3 months, it showed a slight decrease in the presence of periodontal pathogens, which was only significant in the detection of Tf (*p* = 0.031) between the baseline record and six months.

In the test group, with the exception of Aa, the rest of the periodontal pathogens had significant reductions at 2 weeks, and these significant reductions were maintained at three months in Td (*p* = 0.031) and Tf (*p* = 0.008).

After the new treatment and in the 6-month control, a significant reduction was still present in Td (*p* = 0.016) and Tf (*p* = 0.008), and it was also observed in Pg (*p* = 0.031) compared with the baseline record.

When comparing both sides, all the controls presented a greater decrease in the periodontal pathogens in favor of the test group; however, no statistical associations were found, except for Td (*p* = 0.031).

## 4. Discussion

Periodontitis is one of the most prevalent diseases that can lead to tooth loss, with psychosocial consequences for patients [20]. Mechanical treatment by scaling and root planing is a basic element for its control. However, this treatment is often not sufficient, which is why researchers are investigating other adjuvant therapies.

Thus, local antibiotics can be a useful alternative, as they do not depend on patient compliance and allow the drug to be applied directly to the area to be treated, penetrating to the bottom of the periodontal pocket and increasing the action of the scaling and root planing.

The local use of piperacillin–tazobactam differs from other types of products in its ability to gelify after application and its insolubility in liquids; thus, it protects against the entry of new bacteria and prevents the colonization of the tooth surfaces.

In the evolution of periodontitis, the most relevant clinical parameters are the CAL and PPD, which, in this research, noted significant improvements in both groups, which were greater at 3 and 6 months, after the combined treatment (SRP and the application of piperacillin–tazobactam (100/12.5 mg)).

Other studies have used other local antibiotics in similar models, such as macrolides. In a meta-analysis, Bashir et al. [13] noted additional mean reductions in the PPD (1.01 mm at 3 months, and 1.20 mm at 6 months) using a clarithromycin gel; these results are similar to those found in this study. In relation to the CAL, the additional mean gains were 0.56 mm at 3 months and 0.83 mm at 6 months, which are lower values than those observed in this study.

Some studies have also documented the use of tetracyclines. In a multicenter clinical trial conducted by Park et al. [21], the authors analyzed the efficacy of minocycline gel in one group and minocycline–metronidazole gel in another, recording the probing pocket depth, plaque index, and bleeding on probing at the baseline and 3 months later. The mean reduction in the PPD in the minocycline group was 1.88 ± 1.50 mm. The mean reduction in the group with the combination of both antibiotics was 1.95 ± 1.28 mm. The plaque index decreased by a mean of 0.54 ± 0.54, associated with minocycline–metronidazole, and by 0.42 ± 0.54 with the application of only minocycline. The bleeding on probing was reduced by 0.51 ± 0.32 in the minocycline–metronidazole group. The reduction using minocycline alone was 0.50 ± 0.34. The results are comparable to those obtained with the piperacillin–tazobactam association, which was 0.46 ± 0.04 for the plaque index, and 0.40 ± 0.04 for the gingival index.

The favorable response to minocycline is also supported by the systematic review by Tan et al. [14] in 2021, who included nine studies to assess the efficacy of repeated applications of local tetracyclines along with scaling and root planing. At 6 months, with the doxycycline gel, the PPD was reduced by 1.20 ± 1.67 mm, and the CAL was reduced by 0.90 ± 2.26 mm. The results of the application of the minocycline gel were greater, with the PPD reduced by 1, 76 ± 0.63 mm and the CAL reduced by 1.56 ± 0.54 mm.

When comparing these results with those obtained with piperacillin–tazobactam, we observed a better CAL result in this investigation.

The meta-analysis carried out by Yusri et al. [22] produced similar results, with substantial differences at 6 months in the combined treatments with local antimicrobials (minocycline, metronidazole, and doxycycline), and the obtainment of a clinical-attachment-level gain of 0.61 mm (*p* = 0.03), a reduction in the probing pocket depth of 0.41 mm (*p* = 0.04), a reduction in the bleeding on probing of 28.47% (*p* < 0.001), and a mean reduction in the gingival index between 6 and 12 months of 0.27 (*p* = 0.01).

After the application of doxycycline gel as an adjuvant to mechanical periodontal treatment (SRP + doxycycline), Trajano et al. [9] evaluated the plaque index and bleeding on probing at one and two months, compared with the control group (SRP). The bleeding on probing in the SRP + doxycycline group was reduced by 80% between the baseline and control at 2 months, versus 75% in the SRP group alone. The visible plaque index was reduced by 71% in the two groups at the 2-month follow-up. Although both groups achieved significant reductions (*p* < 0.05) in these parameters with respect to the baseline measurement, when comparing the groups with each other, we did not observe any statistically significant differences.

Researchers have also investigated the beneficial effect of local antibiotic therapy in patients with dental implants who, during their evolution, presented peri-implantitis with clinical consequences that were similar to periodontitis.

In a meta-analysis, Toledano et al. [23] included twelve studies (365 patients and 463 implants). After the treatment of peri-implantitis, the probing pocket depth was reduced by 1.40 mm. When the researchers applied local antibiotics (minocycline gel and doxycycline gel), they obtained a reduction in the probing pocket depth of 0.30 mm more than in the control group. The bleeding on probing reached an odds-ratio value of 1.82, which indicates that the probability of bleeding is almost higher when topical antibiotics are not administered.

In the study published by González-Regueiro et al. [24] on the combined efficacy of piperacillin–tazobactam with implantoplasty in 43 patients with peri-implantitis, the authors found that, at one year of follow-up, the probing pocket depth and bleeding and suppuration on probing were significantly reduced, producing a mean bone regeneration of the defect of 2.64 mm ± 1.59 (*p* < 0.001).

Another noteworthy aspect of this research was the use of a new system for the detection of periodontopathogens, which is simple and fast compared with the diagnosis by qPCR.

This system can be used to qualitatively evaluate the presence of five bacteria related to periodontitis, which may represent a certain limitation. However, Arweiler et al. [19] carried out a prospective single-center case–control study to assess its sensitivity and specificity. They examined a total of 125 patients, of which 100 had periodontitis and 25 were periodontally healthy patients, with both the PerioPOC^®^ system and qPCR. The specific clinical limits of detection (LoD) for each bacterium were as follows: 1.2 × 10^4^ colony-forming units (CFUs) for Td and Tf; 2.5 × 10^4^ CFUs for Pg; 5.3 × 10^3^ CFUs for Pi; 5, 8 × 10^4^ CFUs for Aa. Based on this maximum potential for positive detections, the sensitivities of PerioPOC^®^ compared with qPCR in this study were: Td: 91.3%; Tf: 86.3%; Pg: 83.8%; Pi: 85.7%; Aa: 100%. Regarding the clinical diagnosis, sensitivities of 87.82% and 94% were reached for the PerioPOC^®^ system and qPCR, respectively. The specificities for both methods were 100%.

Recently, the same working group published a continuation of this study [25]. Of the 100 patients diagnosed with periodontitis, 74 were treated with nonsurgical periodontal treatment, with the addition of a local (doxycycline) or systemic (amoxicillin/metronidazole, azithromycin, or clindamycin) antibiotic in the other groups. Subsequently, the researchers performed the PerioPOC^®^ test again as qPCR in a follow-up period of between 14 and 24 months. In all patients, the five studied periodontal pathogens could not be completely eliminated. Using the PerioPOC^®^ test, the mean elimination of bacteria varied from 59.13% for the control group (SRP) to 90% for the SRP + amoxicillin–metronidazole group. With qPCR, these values were 29.6% and 30%, respectively. Only the Aa bacteria were 100% eliminated in the SRP + amoxicillin–metronidazole group using the PerioPOC^®^ test, this reduction being 80% when the researchers used qPCR. Both diagnostic tests agreed 98.7% for the detection of Aa, 74.3% for the detection of Pg, 78.4% for the detection of Pi, 73% for the detection of Td, and 48.7% for the detection of Tf. Neither the conventional treatment nor the additional use of antibiotics could completely eliminate the presence of the studied pathogens or prevent the reinfection of the periodontal pockets.

In the present study, we performed the controls and measurements at baseline, 15 days, and 3 and 6 months. According to the obtained results, the greatest improvement in the studied clinical parameters occurred in the time interval from baseline to the 15-day control, after applying the treatment for the first time. In addition, the application of the antibiotic produced a more favorable evolution on the test side. Although we also observed an improvement at 6 months, after repeating the therapy at 3 months, its magnitude was less than that obtained in the first control. However, when comparing both sides of the treatment, we more frequently observed substantial differences between them at 3 and 6 months, which indicates that its application should be repeated to achieve its beneficial effects.

In all patients, we obtained more cases of the presence of the bacteria in the control group and of its absence in the test group. Given the descriptive results obtained, and after observing the behavior of all the microorganisms, we concluded that the application of the local antibiotic as an adjunct to scaling and root planing reduces the presence of bacteria. However, for the statistical test, we analyzed specific bacteria individually. To obtain a significance of *p* < 0.05, more presences are needed on the control side. For this reason, in both groups, we only obtained a significant difference in *Treponema denticola* at the 3-month visit.

Despite these beneficial effects, the treatment of periodontitis remains a challenge, in which other complementary therapies such as the application of prebiotic and postbiotic gels could be a clear alternative, in order to limit the use of antibiotics [26].

The search for alternatives must continue, and further efforts must be made to find optimal treatment protocols for all possible clinical conditions

## 5. Conclusions

The local use of piperacillin–tazobactam is poorly documented in the treatment of periodontitis. According to the results of this research, the use of piperacillin–tazobactam achieved clinical improvements compared to conventional scaling and root planing treatment.

Microbiologically, its application produces a clear substantial decrease in the presence of periodontal pathogens. However, these results are not maintained over time, so a more persistent local application is necessary.

Within the limitations of the present study, further randomized controlled trials of longer follow-ups must be made to confirm these findings.

Nevertheless, to limit the use and potential overuse of antibiotics, the search for novel procedures must continue, and further efforts must be made to find optimal treatment protocols that are efficient in removing bacteria without inducing trauma.

Thus, it follows that periodontitis still does not find a single therapy, so new studies should be opened with other forms of treatment that try to reduce the bacterial load, such as the use of postbiotics.

## Figures and Tables

**Figure 1 antibiotics-11-01689-f001:**
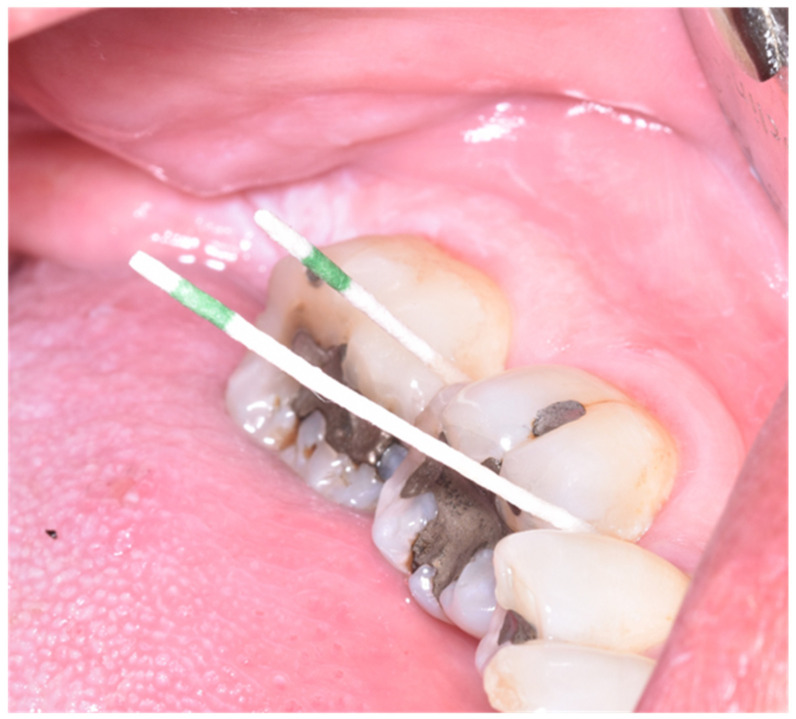
Taking a sample of the subgingival fluid.

**Figure 2 antibiotics-11-01689-f002:**
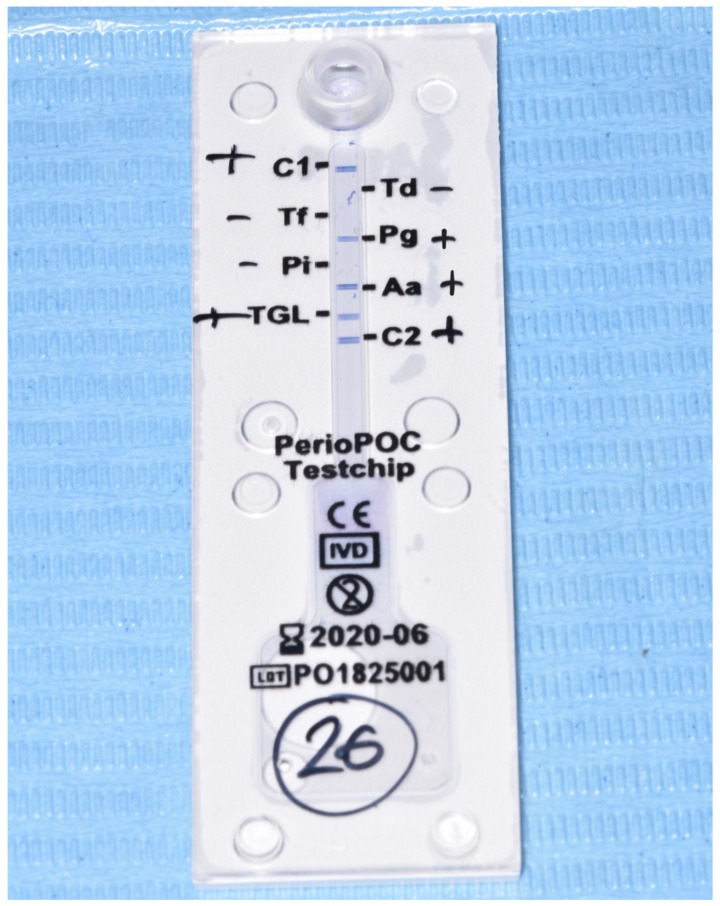
Periodontopathogen determination test (PerioPOC)

**Figure 3 antibiotics-11-01689-f003:**
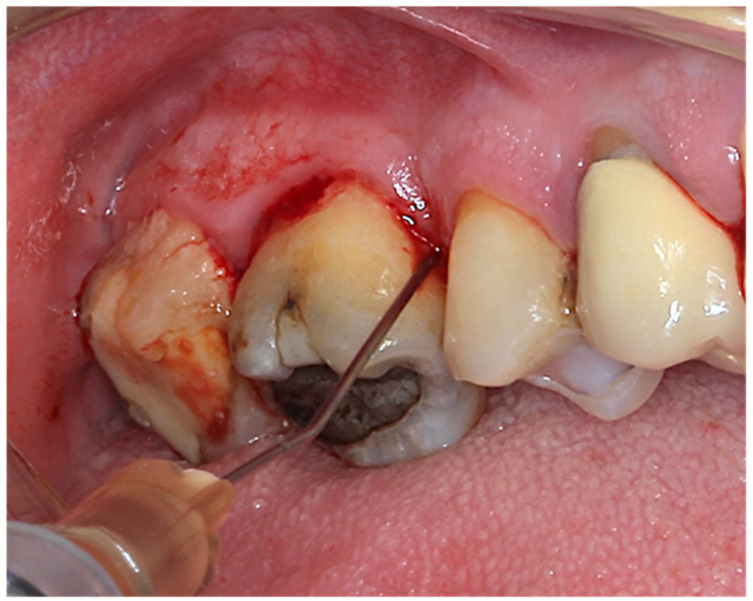
Application of the piperacillin-tazobactam 100/12.5 mg solution (Gelcide).

**Figure 4 antibiotics-11-01689-f004:**
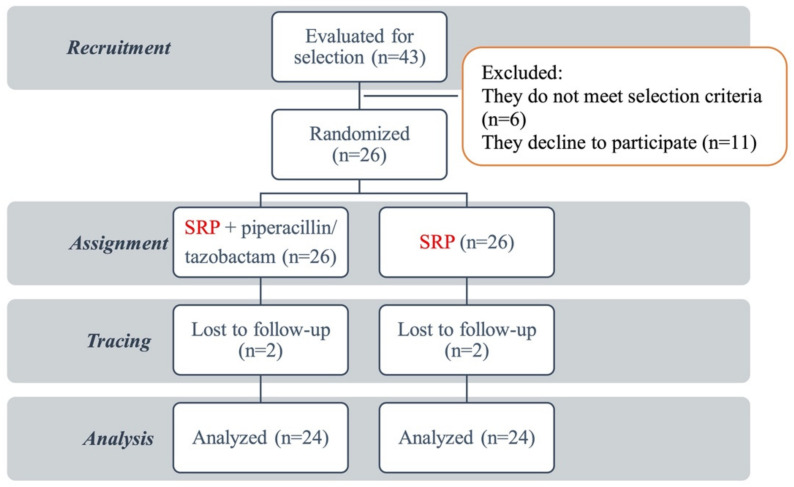
Participant flowchart.

**Table 1 antibiotics-11-01689-t001:** Classification of periodontitis based on stages defined by severity, complexity, and extent and distribution.

Periodontitis Stage		Stage I	Stage II	Stage III	Stage IV
Severity	Interdental CAL at site of greatest loss	1 to 2 mm	3 to 4 mm	≥5 mm	≥5 mm
	Radiographic bone loss	Coronal third (<15%)	Coronal third (15 to 33%)	Extending to mid-third of root and beyond	Extending to mid-third of root and beyond
	Tooth loss	No tooth loss due to periodontitis		Tooth loss due to periodontitis of ≤4 teeth	Tooth loss due to periodontitis of ≥5 teeth
Complexity	Local			In addition to stage II complexity:	In addition to stage III complexity:
		Maximum probing depth ≤4 mm Mostly horizontal bone loss	Maximum probing depth ≤5 mm Mostly horizontal bone loss	Probing depth ≥6 mm Vertical bone loss ≥3 mm Furcation involvement Class II or III Moderate ridge defect	Need for complex rehabilitation due to: Masticatory dysfunction Secondary occlusal trauma (tooth mobility degree ≥2) Severe ridge defect Bite collapse, drifting, flaring Less than 20 remaining teeth (10 opposing pairs)
Extent and distribution	Add to stage as descriptor	For each stage, describe the extent as localized (<30% of teeth involved), generalized, or molar/incisor pattern			

**Table 2 antibiotics-11-01689-t002:** Comparison of CAL, PS, IPL, and IG intergroup and intragroup in the different controls: T0 (baseline), T1 (15 days), T2 (3 months), and T3 (6 months).

Clinical Parameter	Side Control (RAR)	Treatment Side (RAR + Piperacillin/Tazobactam)	
			*p*-Value
Clinical attachment level (CAL) in mm			
T0 (Basal)	6.63 ± 1.44	6.38 ± 1.53	0.398
T1 (15 days)	5.33 ± 1.55 *	4.96 ± 1.33 *	0.131
T2 (3 months)	5.67 ± 1.40 *	4.71 ± 1.27 *	0.001^a^
T3 (6 months)	5 ± 1.25 *	4.25 ± 1.29 *	0.002 ^a^
∆ T0–T3	1.63 ± 0.18	2.13 ± 0.17	
Probing depth (PS) in mm			
T0 (Basal)	4.43 ± 0.84	4.34 ± 0.84	0.553
T1 (15 days)	3.50 ± 0.78 *	3.29 ± 0.78 *	0.139
T2 (3 months)	3.67 ± 0.84 *	3.32 ± 0.8 *	0.024 ^a^
T3 (6 months)	3.47 ± 0.73 *	3.02 ± 0.83 *	0.004 ^a^
∆ T0–T3	0.96 ± 0.14	1.32 ± 0.09	
Plaque Index (IPL)			
T0 (Basal)	1.72 ± 0.64	1.67 ± 0.57	0.069
T1 (15 days)	1.42 ± 0.61 *	1.3 ± 0.56 *	0.000 ^a^
T2 (3 months)	1.46 ± 0.63 *	1.3 ± 0.53 *	0.002 ^a^
T3 (6 months)	1.41 ± 0.61 *	1.21 ± 0.54 *	0.000 ^a^
∆ T0–T3	0.31 ± 0.04	0.46 ± 0.04	
Gingival index (GI)			
T0 (Basal)	1.62 ± 0.65	1.62 ± 0.64	0.992
T1 (15 days)	1.26 ± 0.58 *	1.21 ± 0.56 *	0.234
T2 (3 months)	1.37 ± 0.61 *	1.28 ± 0.59 *	0.105
T3 (6 months)	1.35 ± 0.59 *	1.21 ± 0.6 *	0.034 ^a^
∆ T0–T3	0.27 ± 0.03	0.40 ± 0.04	

Values are presented as mean ± standard deviation. * Intragroup significance with respect to the baseline (*p* < 0.05) and ^a^ Intergroup significance in favor of the treatment side (*p* < 0.05).

**Table 3 antibiotics-11-01689-t003:** Comparison of the presence of intergroup and intragroup bacteria in the different controls: T0 (Basal), T1 (15 days), T2 (3 months), and T3 (6 months) expressed in the number of patients and percentage.

Clinical Parameter	Control Side (RAR)	Treatment Side (RAR + Piperacillin/Tazobactam)	
			*p*-Value
Presence of Aa			
T0 (Basal)	5 (20.8%)	5 (20.8%)	1.000
T1 (15 days)	1 (4.2%)	0 (0%)	1.000
T2 (3 months)	4 (16.7%)	1 (4.2%)	0.250
T3 (6 months)	3 (12.5%)	1 (4.2%)	0.250
∆ T0–T3	2 (8.3%)	4 (16.6%)	
Presence of Pg			
T0 (Basal)	14 (58.3%)	14 (58.3%)	1.000
T1 (15 days)	6 (25%) *	4 (16.7%) *	0.500
T2 (3 months)	13 (54.2%)	10 (41.7%)	0.250
T3 (6 months)	11 (45.8%)	8 (33.3%) *	0.250
∆ T0–T3	3 (12.5%)	6 (25%)	
Presence of Pi			
T0 (Basal)	15 (62.5%)	15 (62.5%)	1.000
T1 (15 days)	9 (37.5%) *	5 (20.8%) *	0.125
T2 (3 months)	15 (62.5%)	13 (54.2%)	0.500
T3 (6 months)	14 (58.3%)	11 (45.8%)	0.250
∆ T0–T3	1 (4.2%)	4 (16.7%)	
Presence of Td			
T0 (Basal)	14 (58.3%)	14 (58.3)	1.000
T1 (15 days)	6 (25%) *	4 (16.7) *	0.500
T2 (3 months)	14 (58.3%)	8 (33.3) *	0.031 ^a^
T3 (6 months)	11 (45.8%)	7 (29.2) *	0.125
∆ T0–T3	3 (12.5%)	7 (29.1%)	
Presence of Tf			
T0 (Basal)	9 (37.5%)	9 (37.5%)	1.000
T1 (15 days)	1 (4.2%) *	0 (0%) *	1.000
T2 (3 months)	5 (20.8%)	1 (4.2%) *	0.219
T3 (6 months)	3 (12.5%) *	1 (4.2%) *	0.625
∆ T0–T3	6 (25%)	8 (33.3%)	

* Intragroup significance with respect to baseline (*p* < 0.05) and ^a^ Intergroup significance in favor of the treatment side (*p* < 0.05).

## Data Availability

The databases used and/or analyzed during the current study are available from the corresponding author upon reasonable request.

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
