# Peer review of "Piperacillin–Tazobactam as an Adjuvant in the Mechanical Treatment of Patients with Periodontitis: A Randomized Clinical Study"

_antibiotics, 2022, doi:10.3390/antibiotics11121689_

Round 1

Reviewer 1 Report

The paper reports the possible use of piperacilline and tazobactam in treatment of periodontitis.

The paper present in their paper a "summary" and an "abstract": this seems to be a double abstract although the concepts reported are different: please re-built a single abstract.

The topic is not innovative in the field and should introduce in the paper neew concepts of antibiotics administration as well as different types of antibiotics and different use of local antibiotics:

1) in the introduction section please modify the following phrase "Given this improvement, researchers are investigating the use of local antibiotics
with the aim of reducing the probing pocket depth and increasing the clinical attachment level compared with individual mechanical treatment"

with the following

"..Given this improvement, and during a worldwide period of time of reduction of use of systemic oral antibiotics, researchers are investigating the use of local antibioticswith the aim of reducing the probing pocket depth and increasing the clinical attachment level compared with individual mechanical treatment"

please cite the following articles

Chisci G, Hatia A. Antibiotics in orthognathic surgery and postoperative infections. Int J Oral Maxillofac Surg. 2022 Aug 27:S0901-5027(22)00320-4. doi: 10.1016/j.ijom.2022.08.008. Epub ahead of print. PMID: 36041952.   D'Ambrosio F, Di Spirito F, De Caro F, Lanza A, Passarella D, Sbordone L. Adherence to Antibiotic Prescription of Dental Patients: The Other Side of the Antimicrobial Resistance. Healthcare (Basel). 2022 Aug 27;10(9):1636. doi: 10.3390/healthcare10091636. PMID: 36141247; PMCID: PMC9498878.  

2)in the introduction section please modify the following phrase "Researchers have used different presentations of different drugs, such as metronida-zole gel, tetracycline fibers, doxycycline gel, minocycline gel or microspheres, azithromy-cin gel, and clarithromycin gel"

into the following

Researchers have used different administrations of different antibiotics, such as metronida-zole gel, tetracycline fibers, doxycycline gel, minocycline gel or microspheres, azithromy-cin gel, and clarithromycin gel in different fields of dentistry

please cite the following paper

Busa A, Parrini S, Chisci G, Pozzi T, Burgassi S, Capuano A. Local versus systemic antibiotics effectiveness: a comparative study of postoperative oral disability in lower third molar surgery. J Craniofac Surg. 2014;25(2):708-9. doi: 10.1097/SCS.0000000000000431. PMID: 24469372.     The author report in their abstract and in the paper the following concept "In conclusion, the local use of piperacillintazobactam produced clinical
improvements that were similar or even superior to those of other local antibiotics" this is not supported by the results, as the local use of these antibiotics produced improvement superior to the control group, or if superior compared to others paper this should be specified as in the abstract as in the paper.

Author Response

Dear reviewer,

We are really thankful to you due to the interest you have shown about our manuscript. We read carefully all your advice and remarks and proceeded to make any necessary changes in order to improve it and achieve accordingly a higher quality article.

We really appreciate the time invested in analyse our work.

The list of changes suggested by you have been done, so we proceed to explain your remarks.

  1. The paper present in their paper a "summary" and an "abstract": this seems to be a double abstract although the concepts reported are different: please re-built a single abstract: Thank you for your suggestion, accordingly, we have rebuilt the abstract,
  2. In the introduction section please modify the following phrase "Given this improvement, researchers are investigating the use of local antibiotics
    with the aim of reducing the probing pocket depth and increasing the clinical attachment level compared with individual mechanical treatment". Thank you. So, we have done.
  3. Please cite the following articles:

. Chisci G, Hatia A. Antibiotics in orthognathic surgery and postoperative infections. Int J Oral Maxillofac Surg. 2022 Aug 27: S0901-5027(22)00320-4. doi: 10.1016/j.ijom.2022.08.008. Epub ahead of print. PMID: 36041952.  

. D'Ambrosio F, Di Spirito F, De Caro F, Lanza A, Passarella D, Sbordone L. Adherence to Antibiotic Prescription of Dental Patients: The Other Side of the Antimicrobial Resistance. Healthcare (Basel). 2022 Aug 27;10(9):1636. doi: 10.3390/healthcare10091636. PMID: 36141247; PMCID: PMC9498878.  

We have added these references in the introduction as references 5 and 6. Consequently we have changed all the references in the text.

  1. In the introduction section please modify the following phrase "Researchers have used different presentations of different drugs, such as metronidazole gel, tetracycline fibers, doxycycline gel, minocycline gel or microspheres, azithromycin gel, and clarithromycin gel". Thank you. So, we have modified.

  1. Please cite the following paper:

. Busa A, Parrini S, Chisci G, Pozzi T, Burgassi S, Capuano A. Local versus systemic antibiotics effectiveness: a comparative study of postoperative oral disability in lower third molar surgery. J Craniofac Surg. 2014;25(2):708-9. doi: 10.1097/SCS.0000000000000431. PMID: 24469372.    

We have added these references in the introduction as reference 15 and consequently we have changed all the references in the text.

  1. The author report in their abstract and in the paper the following concept "In conclusion, the local use of piperacillin–tazobactam produced clinical
    improvements that were similar or even superior to those of other local antibiotics" this is not supported by the results, as the local use of these antibiotics produced improvement superior to the control group, or if superior compared to others paper this should be specified as in the abstract as in the paper. Thank you for your recommendation, so we have redrafted the conclusion : “ The local use of piperacillin–tazobactam is poorly documented in the treatment of periodontitis. According to the results in this research, the use of piperacillin–tazobactam achieved clinical improvements compared to conventional scaling and root planing treatment”.

I remain yours faithfully

Reviewer 2 Report

Dear authors, I would like to congratulate with your team for this RCT, for novelty of this topic, proposing an impeccable conducted study and statistical analysis, with interesting results and conclusions. I've reported some question to explain better your work.

In introduction:

- specify what is SRP (I think it means Scaling and Root Planing)

In material and methods:

- All these procedures (supra-gingival tartrectomy and sub-gingival root planing, with the intrasulcular application of Gelcide) were performed without any type of local anesthesia as lidocaine or mepivacaine? Usually, patients affected by periodontitis need local anesthesia before these procedures

- Patients were not classified into 4 stages of periodontitis severity proposed by the 2018 World Work-Shop classification. Don't you think that different severity of disease could affect the proposed topical treatment outcomes?

- The differential presence of various intrasulcular bacteria registered by PerioPoc at baseline could affect dose or administration frequency of piperacillin-tazobactam gel, in order to customize therapy on each patients need?

Good work !  

Wishing you all success, best regards

Author Response

Reviewer 2
Dear reviewer,

We are really thankful to you  due to the interest you have shown about our manuscript. We read carefully all your advices and remarks and proceeded to make any necessary changes in order to improve it and achieve accordingly a higher quality article. 
We really appreciate the time invested in analyse our work. 
The list of changes suggested by you have been done, so we proceed to explain your remarks.
1.    In introduction: specify what is SRP (I think it means Scaling and Root Planing). Exactly, SRP is scaling and root planning, and we have specified in the introduction. 
2.    In material and methods:
.  All these procedures (supra-gingival tartrectomy and sub-gingival root planing, with the intrasulcular application of Gelcide) were performed without any type of local anesthesia as lidocaine or mepivacaine? Usually, patients affected by periodontitis need local anesthesia before these procedures. In fact, all procedures was carried out under local anesthesia with articaine 4% 1.200000. So we have added in the methodology 
. Patients were not classified into 4 stages of periodontitis severity proposed by the 2018 World Work-Shop classification. Don't you think that different severity of disease could affect the proposed topical treatment outcomes? Than you, we have added in the introduction the Periodontal and Peri-Implant diseases and Conditions Clasiffication . We agree that surely the results would be different depending the stages. 

. The differential presence of various intrasulcular bacteria registered by PerioPoc at baseline could affect dose or administration frequency of piperacillin-tazobactam gel, in order to customize therapy on each patients need? Thank you for your suggestion, we consider that it`s a very interesting line in which we can continue working,  however it has not been evaluated in this study, 

I remain yours faithfully

Reviewer 3 Report

In this study, the authors designed a single-blind split-mouth randomized study to evaluate the effects of the adjuvant piperacillin–tazobactam solution in the mechanical treatment of periodontitis. The results the local use of piperacillin–tazobactam produced clinical improvements that were similar or even superior to those of other local antibiotics. Also, the application of piperacillin–tazobactam produces a clear substantial decrease in the presence of periodontal pathogens. Overall, the manuscript is not well-prepared, and the quality is low. English needs extensive revision. Figure 4 Participants flow chart is confusing. Introducion and Discussion parts are lengthy. Major revision of the manuscript is suggested.

Author Response

 Dear reviewer, 

We are really thankful to you due to the interest you have shown about our manuscript. We read carefully all your advice and remarks and proceeded to make any necessary changes in order to improve it and achieve accordingly a higher quality article.

We really appreciate the time invested in analyse our work.

The list of changes suggested by you have been done, so we proceed to explain your remarks.

In this study, the authors designed a single-blind split-mouth randomized study to evaluate the effects of the adjuvant piperacillin–tazobactam solution in the mechanical treatment of periodontitis. The results the local use of piperacillin–tazobactam produced clinical improvements that were similar or even superior to those of other local antibiotics. Also, the application of piperacillin–tazobactam produces a clear substantial decrease in the presence of periodontal pathogens. Overall, the manuscript is not well-prepared, and the quality is low.

  1. English needs extensive revision. Thank you for your consideration, we apologize for this, but we have used the MPDPI English editing service. We have attached the receipt. Neverthless, we have made changes in oder to improve its readability.
  2. Figure 4 Participants flow chart is confusing. Thank you for your consideration, we have corrected the flow chart.
  3. Introduction and Discussion parts are lengthy. Fixed as directed by all reviewers.

I remain yours faithfully

Reviewer 4 Report

Manuscript of considerable interest for the dental sector.

Before proceeding with the evaluation for publication, it needs a major revision.

Abstract, to highlight the statistical data more

Keywords; add more specifications, these are few

Introduction The reference and description of the new classification of periodontal disease is missing.

Materials and methods; classify patients according to the new classification, and enter how the sample size was calculated.

Results following the modified file, the results are not easily interpretable by ordinary readers who are not experts in scientific research, make them more usable and highlight the requests more

Discussion, being a split mouth study, insert as future objectives, the use of a gel based on postbiotics and probiotics to reduce the bacterial load as already studied by the research group of Prof Scribante, in order to have an approach proactive and reduce the chemical pharmacological action

Conclusion, rephrase it by adding proactive action

Author Response

 Dear reviewer, 

We are really thankful to you due to the interest you have shown about our manuscript. We read carefully all your advice and remarks and proceeded to make any necessary changes in order to improve it and achieve accordingly a higher quality article.

We really appreciate the time invested in analyse our work.

The list of changes suggested by you have been done, so we proceed to explain your remarks.

  1. Abstract, to highlight the statistical data more
  2. Keywords: add more specifications, these are few. Thank you for your suggestion, so we have added new key words.

Introduction The reference and description of the new classification of periodontal disease is missing. We have referenced as 1 the Classification and consequently we have changed all the references in the text. The description has done as a table 1 in the methodology,

  1. Materials and methods; classify patients according to the new classification and enter how the sample size was calculated. We have classified the patients according to 2018 Classification and we have explained in detail how the sample size was calculated: “In order to determine sample size, a pilot study was performed, based on the parameter (probing depth). This study was carried enrolling 5 patients with a split-mouth design, obtaining a mean reduction on probing depth of 1.06 ± 0,78 mm and 0.85 ± 0.66 mm, in test and control side, respectively. Using G Power 3.1 (Dusseldorf, Germany), considering an alpha-type error of 5% and a beta-type error of 5%, the estimation resulted in 20 patients per group.
  2. Results following the modified file, the results are not easily interpretable by ordinary readers who are not experts in scientific research, make them more usable and highlight the requests more. Thank you, so we have done, we hope that this change facilitates their understanding.
  3. Discussion, being a split mouth study, insert as future objectives, the use of a gel based on postbiotics and probiotics to reduce the bacterial load as already studied by the research group of Prof Scribante, in order to have an approach proactive and reduce the chemical pharmacological action. Thank you for your consideration, consequently we have discussed this point: “Despite these beneficial effects, the treatment of periodontitis remains a challenge, in which other complementary therapies such as the application of prebiotic and postbiotic gels could be a clear alternative, in order to limit the use of antibiotics. Also, we have added the reference of Scribante as 26.

  1. Conclusion, rephrase it by adding proactive action. We have rebuilt the conclusion.

I remain yours faithfully

Round 2

Reviewer 1 Report

Accept

Reviewer 2 Report

Dear authors,

with these modifications i think that this paper is suitable for publication. Thank you for listening to my suggestions, which improved the already high scientific value of your work. I suggest you to carry out further studies regarding differential stages of periodontitis or different composition of intrasulcular bacterial flora and their response to your adjuvant therapy. 

Kind Regards

Dr. Giuseppe Barile

Reviewer 3 Report

Accept.

Reviewer 4 Report

The manuscript has been correctly revised, it can be published